# Transferrin-Decorated PLGA Nanoparticles Loaded with an Organoselenium Compound as an Innovative Approach to Sensitize MDR Tumor Cells: An In Vitro Study Using 2D and 3D Cell Models

**DOI:** 10.3390/nano13162306

**Published:** 2023-08-11

**Authors:** Letícia Bueno Macedo, Daniele Rubert Nogueira-Librelotto, Daniela Mathes, Taís Baldissera Pieta, Micheli Mainardi Pillat, Raquel Mello da Rosa, Oscar Endrigo Dorneles Rodrigues, Maria Pilar Vinardell, Clarice Madalena Bueno Rolim

**Affiliations:** 1Programa de Pós-Graduação em Ciências Farmacêuticas, Universidade Federal de Santa Maria, Av. Roraima 1000, Santa Maria 97105-900, Brazil; leticiabuenomacedo@gmail.com (L.B.M.); librelotto.daniele@ufsm.br (D.R.N.-L.); danielamathes@hotmail.com (D.M.); micheli.pillat@ufsm.br (M.M.P.); 2Departamento de Farmácia Industrial, Universidade Federal de Santa Maria, Av. Roraima 1000, Santa Maria 97105-900, Brazil; tais.pieta@acad.ufsm.br; 3Departamento de Microbiologia e Parasitologia, Universidade Federal de Santa Maria, Av. Roraima 1000, Santa Maria 97105-900, Brazil; 4Departamentode Química, Universidade Federal de Santa Maria, Av. Roraima 1000, Santa Maria 97105-900, Brazil; raquelmello.rosa@gmail.com (R.M.d.R.); rodriguesoed@gmail.com (O.E.D.R.); 5Departament de Bioquimica i Fisiologia, Facultat de Farmacia i Ciències de l’Alimentaciò, Universitat de Barcelona, Av. Joan XXIII 27-31, 08028 Barcelona, Spain

**Keywords:** selenium compound, multidrug resistance (MDR), 3D cell culture, active targeted drug delivery, in vitro antitumor activity, cell uptake, flow cytometry

## Abstract

Multidrug resistance (MDR) is the main challenge in cancer treatment. In this sense, we designed transferrin (Tf)-conjugated PLGA nanoparticles (NPs) containing an organoselenium compound as an alternative to enhance the efficacy of cancer therapy and sensitize MDR tumor cells. Cytotoxicity studies were performed on different sensitive tumor cell lines and on an MDR tumor cell line, and the Tf-conjugated NPs presented significantly higher antiproliferative activity than the nontargeted counterparts in all tested cell lines. Due to the promising antitumor activity of the Tf-decorated NPs, further studies were performed using the MDR cells (NCI/ADR-RES cell line) comparatively to one sensitive cell line (HeLa). The cytotoxicity of NPs was evaluated in 3D tumor spheroids and, similarly to the results achieved in the 2D assays, the Tf-conjugated NPs were more effective at reducing the spheroid’s growth. The targeted Tf-NPs were also able to inhibit tumor cell migration, presented a higher cell internalization and induced a greater number of apoptotic events in both cell lines. Therefore, these findings evidenced the advantages of Tf-decorated NPs over the nontargeted counterparts, with the Tf-conjugated NPs containing an organoselenium compound representing a promising drug delivery system to overcome MDR and enhance the efficacy of cancer therapy.

## 1. Introduction

Despite the great advances in cancer treatment, chemotherapy is still the most common anticancer method. However, chemotherapy drugs have shown systemic toxicity and, frequently, cancer cells become progressively unresponsive to these drugs. This multifactorial process, known as multidrug resistance (MDR), is the main outcome responsible for the failure in cancer chemotherapy and cancer recurrence [1,2,3,4]. In this sense, the development of novel approaches, able to improve the effectiveness of antitumor treatment and overcome MDR, are of the utmost importance.

The association of antineoplastic drugs in nanoparticles (NPs) can reduce some drawbacks of chemotherapy, being considered one of the most promising alternatives to increase the effectiveness of cancer treatment [5,6]. Due to the characteristics of the tumor microenvironment, such as enhanced permeability and retention (EPR), these systems can easily penetrate the tumor tissue and accumulate because of attenuated lymphatic drainages [5]. However, the NPs can be modified to direct the drug more precisely to the target site, by active targeting, binding in the NPs molecules able to bind receptors overexpressed at the target tissue [7]. Transferrin (Tf) is a plasma glycoprotein responsible for the transport of iron in the blood stream. Cancer cells present a rapid proliferation rate, increasing iron demand. Therefore, compared to normal cells, transferrin receptors (TfR) are overexpressed in tumor tissue, making Tf a great ligand for antitumor systems [7,8].

Organoselenium compounds have attracted great interest as alternative to cancer treatment because of their promising antitumor activity and ability to prevent metastasis [9,10]. In this sense, a new nucleoside derived from zidovudine, containing selenium (Se), 5′-Se-(phenyl)-3-(amino)-thymidine (ACAT-Se) (Appendix A), was developed by [11] and presented a promising in vitro antitumor activity. Previously, we used this compound to design pH-sensitive NPs [12]. The NPs were prepared using poly(lactic-co-glycolic acid) (PLGA), a biodegradable polymer approved by Food and Drug Administration (FDA) and European Medicine Agency (EMA) [13]; poloxamer as a surfactant, due its great potential to sensitize MDR cells [14]; and 77KL (N^α^,N^ε^-dioctanoyl lysine with an inorganic lithium counterion) as a pH-responsive adjuvant. These pH-sensitive NPs increase ACAT-Se antitumor activity in MCF-7 tumor cells and, associated to the antitumor drug doxorubicin, presented synergism, as well as being able to sensitize MDR cells. Therefore, considering the great antitumor potential of the pH-sensitive NPs, in this study, we proposed the conjugation of Tf to the NP surface, aiming to achieve active targeting and increase the NP selectivity to tumor cells. The Tf-decorated NPs were characterized, and the in vitro drug release profile and scavenging properties were evaluated. The antitumor activity was assessed using different sensitive tumor cells and an MDR tumor cell line. Afterwards, for further studies, we selected the sensitive tumor cell line HeLa and the MDR cell line NCI/ADR-RES, and all the evaluations were performed for the targeted Tf-NPs comparatively to the nonconjugated NPs. The cytotoxicity of the NPs was then assessed using 3D tumor spheroids, and their ability to inhibit tumor cell migration was also measured. Finally, the mechanisms underlying the antitumor activity were evaluated by means of apoptosis rate, cell cycle arrest, cell uptake and intracellular retention.

## 2. Materials and Methods

### 2.1. Reagents and Chemicals

Poly(D,L-lactic-co-glycolic acid) (PLGA, 50:50, 24–38 kDa); sorbitan monooleate (Span 80^®^); poloxamer 407 (Pluronic^®^ F-127); human holo-transferrin; 1-ethyl-3-(3-dimethylaminopropyl) carbodiimine (EDC); 2-2-diphenyl-1-picrylhydrazyl (DPPH); radical,2,2′-azinobis-(3-ethylbenzothiazoline-6-sulfonic acid) (ABTS); rhodamine B, 2,5-diphenyl-3-(4,5-dimethyl-2-thiazolyl) tetrazolium bromide (MTT); phosphate-buffered saline (PBS), trypsin-EDTA solution (0.5 g porcine trypsin and 0.2 g EDTA·4Na per liter of Hank’s Balanced Salt Solution); propidium iodide (PI); RNAse; fetal bovine serum (FBS) and Dulbecco’s Modified Eagle’s Medium (DMEM), supplemented with L-glutamine (584 mg/L); and antibiotic/antimycotic (50 mg/mL gentamicin sulfate and 2 mg/L amphotericin B) were purchased from Sigma–Aldrich (São Paulo, Brazil). An FITC Annexin V Apoptosis Detection Kit was obtained from BD Pharmingen™ (San Diego, CA, USA). All other solvents and reagents were of analytical grade.

5′-Se-(phenyl)-3-(amino)-thymidine (ACAT-Se) was obtained from the LabSelen-NanoBio (Federal University of Santa Maria, Santa Maria, Brazil). This compound was synthesized and fully characterized as previously described by Rosa et al. [11]. ACAT-Se has a molecular weight of 380.31 g/mol and a Log P of 1.05 (value predicted using Molinspiration online software, https://www.molinspiration.com/), is very slightly soluble in water, and soluble in methanol and DMSO.

### 2.2. Preparation of Nanoparticles

The organoselenium compound ACAT-Se was nanoencapsulated on PLGA NPs (ACAT-Se-PLGA-NPs) as previously reported by Macedo et al. [12]. Firstly, ACAT-Se was added to a solution containing PLGA and Span^®^ 80 in acetone; this organic solution was kept for 20 min under magnetic stirring. Then, the organic solution was quickly poured into an aqueous solution containing Pluronic^®^ F-127 and the pH-sensitive adjuvant 77KL. After 10 min under magnetic stirring (530 rpm), the organic solvent was eliminated by evaporation under reduced pressure (ACAT-Se-PLGA-NPs).

This NP suspension was functionalized with Tf following the methodology used by [15,16], with some modifications. Firstly, 2 mL of the NP suspension and 200 µL of EDC (30 mg/mL in water) were maintained under slight stirring at room temperature for 3 h. Then, 300 µL of the Tf solution (5 mg/mL in water) was added and kept under slight stirring at room temperature for 2 h. Finally, the sample was filtered at 3600 rpm for 40 min using a Centrisart^®^ 100 kDa MWCO centrifugal ultrafiltration unit (Sulpeco, Sigma–Aldrich, São Paulo, Brazil). The NP suspension (Tf-ACAT-Se-PLGA-NPs) was collected and the volume was adjusted to 2 mL with ultrapure water.

An NP suspension containing rhodamine B (100 µg/mL) was prepared in the same conditions for the cell uptake studies, and the evaluations of intracellular retention.

### 2.3. Characterization of Nanoparticles

The mean hydrodynamic diameter and the polydispersity index (PDI) were determined via dynamic light scattering (DLS) using a Malvern Zetasizer ZS (Malvern Instruments, Malvern, UK). The NP suspensions were diluted in water (1:500, *v*/*v*), and each measurement was performed using at least three sets of ten runs. The same equipment was used to evaluate the zeta potential (ZP), via electrophoretic mobility. For these measurements, the samples were diluted (1:500, *v*/*v*) in 10 mM NaCl aqueous solution. The pH was determined at room temperature using a calibrated potentiometer (UB-10; Denver Instrument, Bohemia, NY, USA).

NP morphology was assessed using scanning electron microscopy (SEM) (JEOLJSM 6360, Akishima, Japan). For this procedure, 20 µL of the NP suspension was added to a stub and maintained for 12 h at room temperature. Then, the stub was coated with gold under reduced pressure and the samples were analyzed using a potential of 10 kV.

The drug content was determined following the extraction procedure and RP-LC method previously described [12].

The rate of Tf conjugation was evaluated using a commercial kit (BioRad^®^ assay, Hercules, CA, USA) based on the Bradford dye-binding procedure [17]. Firstly, 50 µL of the NP suspension and 50 µL of formic acid were maintained under slight stirring at 1000 rpm for 1 min. The BioRad solution was added (1.5 mL) and the absorbance was measured at 595 nm using a UV-Vis spectrophotometer (UV-1800, Shimadzu, Kyoto, Japan).

### 2.4. In Vitro Release Studies

The release of ACAT-Se from the Tf-conjugated NPs was evaluated using the dialysis method. An aliquot of the NP suspension (500 µL) was placed in a dialysis bag (Sigma–Aldrich, São Paulo, Brazil 14,000 MWCO), sealed, and soaked in 50 mL of phosphate buffer solution (pH 7.4) at 37 °C and kept under constant shaking (150 rpm) for 24 h. At predetermined time intervals, an aliquot of the release medium was collected and replaced with an equivalent amount of fresh release medium to maintain the sink conditions. The amount of ACAT-Se released was determined using the previously published RP-LC method [12]. The release of free ACAT-Se (methanolic solution) was also evaluated, for comparison purposes.

The ACAT-Se release mechanism was studied using the Korsmeyer Peppas equation, as follows:(1)$${kt}^{n}=\frac{Mt}{M\infty }$$
where k considers the geometric characteristics of the system, n gives the information about the diffusional release mechanism of a drug from a polymeric device, and Mt and M∞ are absolute values of drug released at time t and infinite [18,19].

### 2.5. In Vitro Antioxidant Activity

The DPPH and ABTS assays were applied to evaluate the potential antioxidant of the Tf-ACAT-Se-PLGA-NPs, according to the reported literature methods [20,21]. The samples (75 µL) were diluted at the concentrations of 25, 50, 100, 200 and 300 µg/mL and added to 150µL of 50 mM DPPH in methanol. After 30 min of incubation in the dark, the absorbance was measured at 550 nm using a Multiskan FC microplate reader (Thermo Fisher Scientific, Shanghai, China) (Sample). The ABTS solution was prepared via the reaction of 5 mL of 7 mM ABTS in water with 88 µL of 140 mM sodium persulfate. After 12 h in the dark, at room temperature, this solution was diluted in a 10 mM phosphate solution at pH 7.0 to obtain 42.7 µM of ABTS in the final solution. The samples were diluted following the same procedure used for DPPH assay and after the incubation, the absorbance was measured at 734 nm. In both assays, the negative control (Control) was assessed by mixing the DPPH or ABTS solutions with 75 µL of water and the interference of NPs turbidity (blank) was determined by preparing sample solutions with 150 µL of methanol or water instead of the DPPH or ABTS solutions, respectively. Percent of scavenging activity was calculated using the following equation:(2)$$\%\;scavenging\;activity=100-\left(\frac{\left(Sample\right)-\left(Blank\right)\times 100}{\left(Control\right)}\right)$$

### 2.6. Protein Corona Study

This test was used to study if the NPs suffer protein aggregation in the in vitro cytotoxicity assay medium and to predict its behavior in in vivo conditions. The particle size of the NPs was determined after their dilution (60 µg/mL) in cell culture medium (DMEM 5% FBS) or in plasma (1:1, *v*/*v*) [22]. The measurements were performed using a Malvern Zetasizer ZS (Malvern Instruments, Malvern, UK), determined immediately after dilution and after 24 and 72 h of incubation at 37 °C. For comparison purposes, the particle size of the NPs diluted in water was assessed, under the same conditions.

### 2.7. Biological Safety Profile of the NPs: In Vitro Biocompatibility Studies

The biocompatibility of the NPs was evaluated via the hemolysis assay and the assessment of the cytotoxic potential on human mononuclear cells of peripheral blood (PBMCs). Erythrocytes and PBMCs were isolated from human blood, obtained from healthy volunteer donors invited according to the guidelines established by the Ethics Committee in Research, from the Federal University of Santa Maria, Brazil (protocol CAAE 44017921.3.0000.5346). Red blood cells (RBCs) were isolated at a cell density of 8 × 10^9^ cell/mL, as previously reported [23]. The NP suspension at concentrations of 150, 225 and 300 µg/mL were incubated with 25 µL of the erythrocyte suspension for 5 h under gentle shaking. After, the samples were centrifuged at 10,000 rpm for 5 min to stop the reaction. Negative and positive controls were obtained via the incubation of the erythrocyte suspension with PBS at pH 7.4 or water, respectively. The supernatants were placed in 96-well plates and the absorbance of the hemoglobin released was measured at 550 nm using a microplate reader Multiskan FC (Thermo Fisher Scientific, Shanghai, China).

The mononuclear cells were isolated via density gradient centrifugation using Histopaque^®^-1077 (Sigma–Aldrich, São Paulo, Brazil) and cultured in RPMI-1640 medium supplemented with 10% (*v*/*v*) FBS. The PBMCs were seeded in a 96-well plate at 5 × 10^5^ cells/mL, and treated for 24 h with the NPs at 60, 40, 20, 10, 5 and 0.5 µg/mL. The cell viability was determined using the MTT assay. The cells were incubated with MTT (0.5 mg/mL) for 3 h at 37 °C with 5% CO_2_. Finally, absorbance was measured at 550 nm using a Multiskan FC microplate reader (Thermo Fisher Scientific, Shanghai, China).

### 2.8. Cell Lines and Culture Conditions

The tumor cell lines A375 (human melanoma), HeLa (human epithelial cervical cancer), MCF-7 (human breast cancer) and U-87 (human glioblastoma) were cultured in DMEM medium (4.5 g/L glucose) supplemented with 10% (*v*/*v*) FBS, at 37 °C with 5% CO_2_. The multidrug-resistant (MDR) cell line NCI/ADR-RES (human ovarian cancer cells) was kindly donated by Dr. Antoni Benito from the University of Girona (Girona, Spain) and cultured under the same conditions using 1 µg/mL of doxorubicin (DOX) in the culture medium. The cells were routinely grown into 75 cm^2^ cultivation flasks at 37 °C in a humidified atmosphere of 5% CO_2_ until they reached approximately 80% confluence.

### 2.9. In Vitro Antitumor Activity Screening Using 2D Cell Models

All tumor cells lines were seeded in 96-well plates and incubated at 37 °C in a humidified atmosphere of 5% CO_2_ for 24 h. After, the cells were treated with the Tf-ACAT-Se-PLGA-NPs, ACAT-Se-PLGA-NPs or free ACAT-Se (at concentrations of 0.5, 5, 10, 20, 40 and 60 µg/mL of ACAT-Se). The antitumor drugs methotrexate (50 µg/mL) and doxorubicin (10 µg/mL) were also assessed, for comparison purposes. The cells were treated for 24 or 72 h, and cell viability was determined using the MTT assay. The cells were incubated for 3 h at 37 °C with 5% CO_2_ with 0.5 mg/mL of the MTT solution in DMEM without FBS. Then, this solution was replaced by 100 µL of DMSO and the absorbance was measured at 550 nm using a microplate reader Multiskan FC (Thermo Fisher Scientific, Shanghai, China). The results are expressed as percentage of viability with regard to untreated control cells, and the IC_50_ (concentration required to cause 50% death of the cell population) of the treatments were calculated by the dose–response curves plotted for each cell line. Moreover, the selectivity of the Tf-decorated NPs by the tumor cells was defined by the selectivity index (SI), which was calculated as follows:(3)$$Selectivity\;index\;(SI)=\frac{{IC}_{50}\;in\;nontumor\;cell\;line}{{IC}_{50}\;in\;tumor\;cell\;line}$$

### 2.10. In Vitro Antitumor Activity Assays Using 3D Cell Models

The HeLa and NCI/ADR-RES spheroids were obtained via the hanging-drop method. Drops of 20 µL of cell suspensions (HeLa at 1.5 × 10^5^ cell/mL and NCI/ADR-RES at 3 × 10^5^ cell/mL) were placed in the lid of a Petri dish, and the lids were inverted over dishes containing 10 mL of sterile ultrapure water and kept at 37 °C in a humidified atmosphere of 5% CO_2_ for the formation of the cell aggregate. Then, each drop was transferred to a well of 96-well plate precoated with agarose containing 180 µL of DMEM 10% FBS (HeLa) or DMEM 10% FBS + DOX 1 µg/mL (NCI/ADR-RES). Afterwards, the plates were incubated for the formation of the tumor spheroids. Then, the treatments at concentrations of 20, 40 and 60 µg/mL of ACAT-Se for HeLa, and 60, 90 and 120 µg/mL of ACAT-Se for NCI/ADR-RES, were applied and kept for 12 days. The spheroids were photographed during the treatment, and their areas were determined using ImageJ/Fiji software (ImageJ 1.53k, National Institute of Health, Bethesda, MD, USA). Results were expressed as spheroid area percentage (%) determined in comparison to day 0, which was set as 100%, calculated using the following equation:(4)$$Spheroids\;area\;percentage\;(\%)=\left(\frac{spheroids\;area\;t\times 100}{spheroids\;area\;t0}\right)$$

### 2.11. Cell Migration

The tumor cells were seeded on 24-well plates at 1 × 10^5^ and 1.5 × 10^5^ cells per well for HeLa and NCI/ADR-RES, respectively. The plates were incubated at 37 °C in a humidified atmosphere of 5% CO_2_ for 24 h for the formation of the monolayer. Then, the monolayer was scratched using a sterile 200 µL micropipette tip, the medium was removed, and the cells were rinsed twice with cold PBS. The treatments were diluted in DMEM 2% FBS at 20 µg/mL, and the scratched area was photographed during a 48 h period. Untreated cells were used as control. The wound closure was measured at 10 random points between the two wound fronts of each image using ImageJ/Fiji software (ImageJ 1.53k, National Institute of Health, Bethesda, MD, USA) [24]. Results were expressed as cell migration (%) determined in comparison to mean wound closure at time 0, which was set as 100%, calculated using the following equation:(5)$$Cell\;migration\left(\%\right)=\left(\frac{(mean\;wound\;clousure\;t0\;-mean\;wound\;clousure\;t)\times 100}{mean\;wound\;clousure\;t0}\right)$$

### 2.12. Cell Uptake Studies

The HeLa and NCI/ADR-RES tumor cells were seeded in 6-well plates at 6.5 × 10^4^ and 1 × 10^5^ cell/mL, respectively. Both Tf-conjugated and nontargeted rhodamine B-loaded NPs were incubated with the cells at 3 µg/mL, for 4 h at 37 °C. Afterwards, the cells were washed with PBS, harvested with trypsin and transferred to falcon tubes. Then, the cells were washed two times with PBS and resuspended in 200 µL of PBS. The rate of NP cell internalization was assessed via flow cytometry using a BD Accuri™ C6 flow cytometer (BD Biosciences, San Jose, CA, USA) at 50,000 gated events.

### 2.13. Intracellular Retention

The intracellular retention of NPs was evaluated via flow cytometry using the rhodamine B-loaded NPs. The tumor cell lines HeLa and NCI/ADR-RES at 6.5 × 10^4^ and 1 × 10^5^ cell/mL, respectively, were seeded in 6-well plates. After 24 h, the treatments (3 µg/mL) were applied for 2 h. Then, the treatment-containing medium was withdrawn and cells were kept with fresh DMEM for 1, 2 or 4 h. The plates were washed two times with PBS, and then the cells were harvested with trypsin and resuspended in 200 µL of PBS.

### 2.14. Determination of Apoptosis Rate

The tumor cells were seeded in 6-well plates, at 1 × 10^5^ cells/mL, and incubated at 37 °C in a humidified atmosphere of 5% CO_2_. Then, the cells were treated with free ACAT-Se, ACAT-Se-PLGA-NPs or Tf-ACAT-Se-PLGA-NPs at 30 µg/mL. After 24 h, cells were harvested, washed two times with PBS and resuspended in binding buffer 1×, followed by incubation with 5 µL of Annexin-V FITC and 5 µL of PI for 15 min. The apoptosis rate was then measured via flow cytometry (BD AccuriC6, BD Bioscience).

### 2.15. Cell Cycle Analysis

The HeLa and MDR cells were cultured at 1 × 10^5^ cells/mL in 6-well plates. The cells were treated with free ACAT-Se, ACAT-Se-PLGA-NPs or Tf-ACAT-Se-PLGA-NPs at 30 µg/mL, for 24 h. Then, the cells were harvested, washed two times with 500 µL of cold PBS, fixed with 4.5 mL of ice-cold ethanol (70%) and kept at −20 °C. Afterwards, the cells were resuspended in DNA extraction buffer and incubated at 37 °C for 30 min. Then, the samples were centrifuged and resuspended with 500 µL of a solution containing 20 μg/mL PI, 200 μg/mL RNAse and 0.1% Triton X-100 in PBS. The cell cycle analyses were carried out using flow cytometry (BD AccuriC6, BD Bioscience).

### 2.16. Statistics

Results are shown as mean ± standard error (SE) or mean ± standard deviation (SD), and statistical analyses were performed using one-way analysis of variance (ANOVA) with the Student–Newman–Keuls test for multiple comparisons, using SPSS^®^ software (version 22.0, SPSS Inc., Chicago, IL, USA).

## 3. Results

### 3.1. Characterization of Nanoparticles

The Tf-ACAT-Se-PLGA-NPs presented a particle size of 137 ± 16.66 nm, PDI of 0.127 ± 0.03, ZP of −4.36 ± 0.93 mV and pH of 7.38 ± 0.30. The individual characterization results of three different batches are presented in Appendix A, and the corresponding graphical of particle size and ZP distribution is in Appendix A. Moreover, SEM images showed that the NPs have a spherical morphology and regular surface (Figure 1).

The Tf-ACAT-Se-PLGA-NPs drug content was quantified via the RP-LC method as 1.50 ± 0.14 mg/mL, which corresponds to 78.00% of the theoretical drug content of the primary formulation, the ACAT-Se-PLGA-NPs (1923 µg/mL) [12]. It is worth mentioning that the targeted Tf-conjugated NPs were obtained from this primary formulation.

Finally, Tf was effectively conjugated to the NPs presenting a conjugation rate of 0.745 ± 0.03 mg/mL, corresponding to 99.37%.

### 3.2. In Vitro Release Studies

The in vitro release studies were performed to assess the release behavior of ACAT-Se from Tf-PLGA-NPs. As can be seen in Appendix A, Tf-ACAT-Se-PLGA-NPs presented a controlled release profile, compared to free ACAT-Se. The release percentages after 2 h were 82.67 ± 3.40 and 53.06 ± 3.80 from free ACAT-Se and Tf-ACAT-Se-PLGA-NPs, respectively. The release of ACAT-Se from the NPs shows an initial burst release that can be associated to the compound adsorbed in the NPs’ surface, while the controlled release can be related to the ACAT-Se encapsulated inside the PLGA polymeric matrix. In addition, the Korsmeyer Peppas equation indicates that Fickian diffusion (n = 0.22) is the release mechanism of ACAT-Se from the NPs.

### 3.3. In Vitro Antioxidant Activity

The radical scavenging activity of free ACAT-Se and Tf-ACAT-Se-PLGA-NPs via the DPPH and ABTS assays is presented in Appendix A. In both assays, free ACAT-Se presented low scavenging activity regardless of the concentration (less than 24%). On the other hand, the encapsulation of the organoselenium compound in the targeted NPs increased their antioxidant activity significantly. The scavenging activity of Tf-ACAT-Se-PLGA-NPs was superior to that of free ACAT-Se in all the concentrations tested via the ABTS assay and in the higher concentrations tested via the DPPH assay (*p* < 0.05).

### 3.4. Protein Corona Study

The mean hydrodynamic size of the NPs in the different conditions is represented in Appendix A. The NPs did not present immediate aggregation, as no difference in mean hydrodynamic diameter was evidenced between the conditions in time 0 (110.53, 104.23 and 107.53 nm are the mean hydrodynamic sizes of NPs incubated with water, DMEM 5% FBS and plasma, respectively, *p* > 0.05). Likewise, no increase in mean hydrodynamic size was observed after incubation at 37 °C for 72 h with DMEM 5% FBS or plasma (*p* > 0.05), suggesting that the NPs did not suffer significant adsorption of biomolecules.

### 3.5. In Vitro Biocompatibility Studies

Through the hemolysis assay, the hemoglobin released was quantified and used to evaluate erythrocyte damage. The Tf-ACAT-Se-PLGA-NPs presented a hemolysis rate lower than 3%, even in concentrations as high as 300 µg/mL, and can be considered non-hemolytic in the concentrations tested (Figure 2A). In addition, in the MTT assay, the Tf-decorated NPs, in concentrations lower than 40 µg/mL, displayed negligible cytotoxicity in PBMCs. In contrast, the highest tested concentrations of 40 and 60 µg/mL displayed low and moderate cytotoxicity, respectively (Figure 2B).

### 3.6. In Vitro Antitumor Activity Using 2D Cell Monolayer Assays

The antitumor activity of free ACAT-Se, unloaded-NPs, ACAT-Se-PLGA-NPs and Tf-ACAT-Se-PLGA-NPs are illustrated in Figure 3. The data show that the NPs inhibit the cell proliferation in a time-dependent manner, and the Tf-decorated NPs were more cytotoxic than free ACAT-Se and ACAT-Se-PLGA-NPs, mainly at 40 and 60 µg/mL. On the sensitive tumor cell lines MCF-7, A375 and U-87, free ACAT-Se presented low cytotoxicity, even after 72 h of treatment, showing 62.61, 77.74 and 86.94% cell viability, respectively, at the highest concentration. In the same cell lines, the association of ACAT-Se to the NPs increased the cytotoxicity of the compound, indicating the advantages of its nanoencapsulation. The ACAT-Se-PLGA-NPs resulted in cell viabilities of 22.25, 46.04 and 43.82% after 72 h of treatment in MCF-7, A375 and U-87, respectively. It is worth mentioning that the increase in cytotoxicity was even more significant after the conjugation of Tf to the NPs surface. The 72 h treatment with the Tf-NPs reduced the cell viability to 7.72, 2.47 and 12.01% in MCF-7, A375 and U-87, respectively. On the other hand, in the HeLa cell line, the ACAT-Se-PLGA-NPs and free ACAT-Se presented a similar cytotoxic effect: 60.92 and 58.45, respectively. However, the benefit of the Tf association to the NPs was also evidenced by the considerable decrease in cell viability to 5.33%.

Furthermore, the antitumor activity of the NPs was assessed using an MDR cell line (NCI/ADR-RES). As expected, this cell line presents a higher resistance, with the Tf-decorated NPs being the only treatment that is effectively cytotoxic. The Tf-NPs reduce the cell viability to 23.09% after 72 h of treatment; conversely, free ACAT-Se and ACAT-Se-PLGA-NPs resulted in 91.47 and 78.85% cell viability rates, respectively. In addition, it is worth highlighting that in the MDR cells, the Tf-NPs, in the higher concentrations, were more cytotoxic than the antitumor drugs DOX and MTX, used as positive control in the assay. Likewise, on the sensitive tumor cells, Tf-decorated NPs presented equal or even greater cytotoxicity than that observed for DOX and MTX, mostly at 40 and 60 µg/mL.

Moreover, the IC_50_ and the SI values resulting from 24 h of treatment for Tf-ACAT-Se-PLGA-NPs are presented in Table 1. The IC_50_ values calculated for the tumor cell lines are lower than those obtained for the nontumor cells (PBMCs), indicating the selectivity of Tf-decorated NPs by the tumor cells. This selectivity can be confirmed by the SI values, as Tf-ACAT-Se-PLGA-NPs showed an SI > 1.9.

The IC_50_ values obtained for free ACAT-Se, ACAT-Se-PLGA-NPs and Tf-ACAT-Se-PLGA-NPs after 72h of treatment are shown in Appendix A. The higher antitumor activity of Tf-conjugated NPs is confirmed by the low IC_50_ values obtained for this formulation (between 3.53 and 11.72 µg/mL for sensitive tumor cells and 23.73 µg/mL for MDR cells). In addition, the Tf-ACAT-Se-PLGA-NPs presented lower IC_50_ values than free ACAT-Se and ACAT-Se-PLGA-NPs in all tumor cells. The IC_50_ for unloaded NPs (24 and 72 h), and for free ACAT-Se and ACAT-Se-PLGA-NPs after 24 h, could not be determined due to the low cytotoxicity evidenced by the treatments in these experimental conditions.

### 3.7. In Vitro Antitumor Activity Assays Using 3D Cell Models

The spheroid growth inhibition assays were carried out after treatment with free ACAT-Se, ACAT-Se-PLGA-NPs or Tf-ACAT-Se-PLGA-NPs. In general, the data showed that the NPs, especially the Tf-decorated NPs, presented growth inhibition in a time- and concentration-dependent manner. The reduction in spheroid size after treatment with Tf-conjugated NPs can even be visually noted (Figure 4). Firstly, the HeLa tumor spheroid results (Figure 5) indicated that both ACAT-Se-PLGA-NPs and Tf-ACAT-Se-PLGA-NPs inhibit spheroid growth after 12 days of treatment, at 20 and 40 µg/mL, in comparison to free ACAT-Se and control spheroids (*p* < 0.05). However, the Tf-decorated NPs presented the most effective growth inhibition in HeLa spheroids at the highest concentration, being more cytotoxic than all other treatments after 9 and 12 days. At the end of the assay (day 12th), control HeLa spheroids showed an area of 173.53%, in comparison to day 0, which was set as 100%. HeLa spheroids treated with 60 µg/mL of free ACAT-Se presented a growth rate similar (177.80%) to control spheroids, whereas spheroids treated with ACAT-Se-PLGA-NPs (60 µg/mL) demonstrated a slight reduction in growth rate (131.62%). Conversely, Tf-decorated NPs (60 µg/mL) significantly decreases spheroid growth to 88.21%; therefore, they were able to inhibit 85.32% of the growth rate with regard to the control spheroids.

Likewise, the Tf-conjugated NPs were the treatment that displayed the highest inhibition in NCI/ADR-RES spheroid growth. The Tf-ACAT-Se-PLGA-NPs were more effective than all other treatments from day 7, at 90 µg/mL; and from day 5, at 120 µg/mL. As observed for HeLa spheroids, control NCI/ADR-RES spheroids grew during the assay and, at day 12, presented an area of 132.54%, in comparison to day 0, which was set as 100%. The NCI/ADR-RES spheroids, treated with 120 µg/mL of free ACAT-Se, ACAT-Se-PLGA-NPs and Tf-ACAT-Se-PLGA-NPs, presented, at the end of the assay (day 12), an area of 101.73, 105.40 and 60.93%, respectively, in comparison to their initial area. Thus, the Tf-decorated NPs reduced the spheroid growth by 71.61, 40.80 and 44.48% in comparison to the control, free ACAT-Se and ACAT-Se-PLGA-NPs, respectively.

### 3.8. Cell Migration

The wound-healing assay was used to determine if the Tf-decorated NPs were able to inhibit tumor cell migration. As shown in Figure 6, free ACAT-Se did not present an inhibitory effect on cell migration compared to the control (*p* > 0.05), in both cell lines. Conversely, the ACAT-Se-PLGA-NPs, after 48h, reduced the HeLa and NCI/ADR-RES migration more effectively than untreated and free ACAT-Se-treated cells. Moreover, Tf-conjugated NPs were the most effective treatment, being able to inhibit cell migration remarkably in both cell lines. The HeLa cells migrated 76.84, 55.03 and 20.46%, after 48 h, for untreated, ACAT-Se-PLGA-NPs and Tf-ACAT-Se-PLGA-NPs, respectively. Likewise, the untreated MDR cells presented a cell migration of 68.64% after 48 h of treatment, whereas the cells treated with Tf-decorated NPs migrated only 31.86%.

### 3.9. Cell Uptake Studies

The internalization of nontargeted NPs and Tf-conjugated NPs in HeLa and NCI/ADR-RES cells was studied using flow cytometry. In both cell lines, the Tf-conjugated NPs displayed a higher cell uptake than nontargeted NPs, as can be seen in Figure 7. The Tf-decorated NPs were 39.5- and 1.7-fold more internalized than nontargeted NPs in HeLa and NCI/ADR-Res, respectively.

### 3.10. Intracellular Retention

The intracellular retention assay was performed to determine if the amount of drug retention increased using the Tf-decorated NPs (Figure 8). In both cell lines, the Tf-conjugated NPs presented a higher mean fluorescence intensity during all assay, in comparison to nontargeted NPs. In addition, after 4 h, both NPs presented a slight reduction in mean fluorescence intensity; even so, the Tf-decorated NPs exhibit a superior retention.

### 3.11. Determination of Apoptosis Rate

The apoptosis rate in HeLa and NCI/ADR-RES cells was evaluated using the Annexin V-FITC/PI assay, which allowed the differentiation of viable cells, early apoptotic and late apoptotic/necrotic cells. The scatter plots of the cells after the treatments are illustrated in Appendix A. Cells treated with free ACAT-Se and ACAT-Se-PLGA-NPs showed no difference in the number of viable cells and apoptotic events in comparison to control cells, in both HeLa and NCI/ADR-RES cells (Figure 9). Conversely, the Tf-decorated NPs significantly reduced the number of viable cells and increased the number of apoptotic events in comparison to all other treatments, in both cell lines. The HeLa cells treated with Tf-ACAT-Se-PLGA-NPs presented 66.30% of cells in early apoptosis, which was 2.38 and 4.83 higher than ACAT-Se-PLGA-NPs and free ACAT-Se, respectively. Likewise, in the MDR cells, the Tf-conjugated NPs increased the rate of the cells in early apoptosis to 33.45%, against the 11.61 and 9.34% shown by free ACAT-Se and ACAT-Se-PLGA-NPs, respectively.

### 3.12. Cell Cycle Analysis

Flow cytometry was used to study alterations in the cell cycle of HeLa and NCI/ADR-RES cells induced by free ACAT-Se, ACAT-Se-PLGA-NPs and Tf-ACAT-Se-PLGA-NPs. Histograms of cell distribution in the different phases of the cell cycle and the graphics of the cell cycle phase in percentage are shown in Appendix A and Figure 10, respectively. Firstly, free ACAT-Se did not modify the cell cycle in both cell lines. The HeLa cells treated with ACAT-Se-PLGA-NPs showed an increased percentage of cells in the sub-G1 phase. On the other hand, the treatment with Tf-ACAT-Se-PLGA-NPs arrested the cell cycle in sub-G1 (24.35% against 5.39% in control) and G2/M (26.25% against 11.30% in control) phases.

In the MDR cells, the only treatment that induced alterations in the cell cycle was the Tf-decorated NPs, whereby a cell cycle arrest was evidenced in the S (23.45% against 18.35% in control) and G2/M (32.80% against 22.05% in control) phases.

## 4. Discussion

Active targeting is a successful strategy to increase the efficacy of cancer treatment, since it can improve the affinity of NPs to cancer cells [25]. In this sense, Tf is a biomarker which has been associated to nanocarriers to target tumor cells [26,27,28,29]. Our previous data evidenced that pH-sensitive PLGA-NPs encapsulating the organoselenium compound ACAT-Se were cytotoxic to the sensitive tumor cell line MCF-7 and, when associated with DOX, presented synergistic action resulting in the sensitization of NCI/ADR-RES cells [12]. Therefore, we conjugated the glycoprotein Tf to the surface of pH-sensitive NPs, aiming to make them more cytotoxic, especially towards MDR tumor cells, but without the association with DOX.

The Tf was conjugated to the NPs via the carbodiimide coupling reaction. By this process, the terminal –COOH groups of PLGA react with EDC, producing an amine intermediate. Subsequently, the free amino groups of Tf were linked to the polymer, forming the conjugated PLGA-Tf [13,16]. The conjugation of Tf to the NPs was suitable, evidenced by the high conjugation rate obtained (99.37%). The Tf-decorated NPs showed a particle size of 137.02 nm, which is considered adequate to accumulate in tumor tissue due to the EPR effect [30]. In addition, this is the optimal size to avoid being cleared by the liver and spleen system [31]. The NPs showed a spherical morphology in SEM analysis (Figure 1), which can be favorable, since spherical polymeric NPs showed a higher cell uptake than non-spherical NPs [32]. The negative ZP can be assigned to the negative charge of the carboxyl group end chain in PLGA [33], and the low module value to the non-ionic surfactant Pluronic^®^ F-127 used as stabilizer in the NPs [34]. In this case, the stabilization of the formulation occurs via the steric effect [35].

The in vitro release studies (Appendix A) demonstrated that the conjugation of Tf to PLGA-NPs did not change the release of ACAT-Se, as the cumulative release of ACAT-Se after 24 h was 56.3 and 55.67% for the unconjugated [12] and Tf-conjugated NPs, respectively.

As several organoselenium compounds demonstrate great antioxidant activity [36,37], here, we assessed the antioxidant potential of free and nanoencapsulated ACAT-Se (Appendix A). Free ACAT-Se presented low scavenging activity via the DPPH and ABTS assays; conversely, the Tf-decorated ACAT-Se-loaded NPs showed a significant enhancement of the ACAT-Se antioxidant activity in both scavenging activity assays. These results supported those obtained for the pH-dependent NPs [12], suggesting that the Tf conjugation did not suppress the NPs’ antioxidant activity. Moreover, the enhancement of the ACAT-Se antioxidant activity after the nanoencapsulation might be related to an increase in the ACAT-Se stability in the assay medium.

Once dispersed in a biological medium, the NPs can spontaneously suffer the adsorption of proteins and lipids, promoting the formation of a protein corona on their surfaces [38,39]. The formation of a protein corona led to modifications of the NPs’ physiological behavior, affecting their colloidal stability and intracellular uptake, increasing clearance and, consequently, reducing their delivery to the targeted tissue [39,40]. The NPs were incubated with culture medium and plasma and, in both conditions, the NPs did not present an increase in particle size, suggesting that the NPs did not suffer the adsorption of biomolecules (Appendix A). The formation of a protein corona can be affected by different factors, such as the NPs’ surface charge. In this sense, previous studies have evidenced that NPs with a neutral ZP showed fewer opsonization than charged ones [38,41]. Hence, the NPs’ low ZP value could be the reason for the absence of protein corona formation, as well as the presence of Pluronic^®^ F-127, which can be related to a decrease in adsorbed proteins [38]. Even though the protein corona is usually harmful for NPs, its presence may be particularly disadvantageous to actively target NPs, since it might reduce the targeting abilities of NPs [41].

In vivo models were originally developed to investigate the efficacy and toxicity of new formulations; however, some limitations such as the availability of animals, high maintenance costs and ethical aspects, make the development of efficient in vitro assays to predict in vivo toxicity essential [42,43]. The use of RBCs to evaluate NP toxicity is very useful, especially the hemolysis assay, which presents results greatly related to in vivo toxicity studies [43,44]. In the present study, the Tf-decorated NPs showed non-significant hemolysis, suggesting their biocompatibility with RBCs (Figure 2). Moreover, the NPs’ compatibility was assessed via the MTT assay using PBMCs. This cell model was more sensitive, and the conjugated NPs showed low toxicity only up to 40 µg/mL.

Likewise, in vitro cell culture assays have been widely used in drug screening, due to some advantages such as using a small sample amount, as well as simplicity and reproducibility [42]. Therefore, cell-based models were selected to evaluate the antitumor activity of free and nanoencapsulated ACAT-Se using different sensitive tumor cell lines (A375, HeLa, MCF-7 and U-87) and one MDR cell line (NCI/ADR-RES) (Figure 3). Generally speaking, the treatments showed concentration- and time-dependent antitumor activity; moreover, regardless of the cell line, the Tf-decorated NPs were the most cytotoxic treatment, being the only one able to sensitize the MDR cells. Additionally, these NPs showed lower IC_50_ values in the tumor-cell than in the nontumor-cell PBMCs (SI values > 1.9), suggesting that the Tf-decorated NPs present selectivity for the tumor cells (Table 1). As the SI value demonstrates the differential activity of a formulation, the greater the index value is the more promise it holds.

It is well known that traditional cell culture models (2D) have some limitations, as they are not able to mimic cell–cell and cell–extracellular environment interactions. Therefore, cytotoxicity observed in the monolayer cell culture can produce results different from those assessed in vivo [45]. In this context, 3D models have advantages, particularly for the study of anticancer drugs. Tumor spheroids can simulate the dense structures of solid tumors, presenting hypoxia, limited molecular transport, low interstitial pH and metabolic changes. Consequently, tumor spheroids can be considered a more suitable model to predict antitumor effects in vitro than monolayer cultures [42,45]. Therefore, here, we also assessed the cytotoxicity of the nanoformulations using a tumor-spheroid 3D model (Figure 4 and Figure 5). Similar to the results observed in the 2D model, the Tf-conjugated NPs were also the most cytotoxic treatment in the tumor spheroids. In addition, besides the inhibition of spheroid growth, this nanoformulation was able to reduce the spheroid size regardless of their initial size. However, it is worth mentioning that even though the same pattern has been shown in 2D and 3D models, the NPs only presented cytotoxicity in the highest tested doses in the tumor spheroid model. This behavior was especially observed in the MDR spheroids, supporting the statement that 3D models show greater treatment resistance in comparison to 2D models [42]. Tf-conjugated RBC membranes PLGA NPs encapsulating DOX and methylene blue also evidenced that the 3D model is more resistant than the 2D model. In this study, a significant increase in the IC_50_ of the treatments was observed in the tumor spheroid model. In addition, in both 2D and 3D models, the Tf-conjugated NPs were more cytotoxic than the non-conjugated NPs [29]. Similarly, Tf-decorated pH-sensitive PLGA NPs encapsulating DOX were more cytotoxic than the non-conjugated counterparts in MCF-7 and NCI/ADR-RES cultured in a monolayer [26]. Moreover, Tf-conjugated docetaxel-PLGA NPs showed greater antitumor activity in MCF-7 cells than non-conjugated NPs [28]. The higher expression of TfR in tumor cells can be associated to the Tf-decorated NPs great antitumor activity. Moreover, the higher activity of Tf-NPs observed towards the MDR cells may be related to an even greater expression of TfR on metastatic and MDR tumors when compared to their normal counterparts [8,46].

In tumor cells, the ability to migrate is related to tumor invasion, neoangiogenesis and metastasis; in this sense, the development of an anticancer therapy that also presents an antimigratory effect is highly relevant [47,48]. The inhibition of cell migration was assessed via the wound-healing assay (Figure 6), and our data evidenced an increase in the ACAT-Se antimigrating activity after its association to the NPs. Noteworthy is that this activity increased even more when ACAT-Se is encapsulated in Tf-conjugated NPs. Thus, these findings suggest that ACAT-Se loaded in the Tf-decorated NPs can potentially inhibit cell migration and prevent the occurrence of metastasis. Other organoselenium compounds such as selenomethionine, methylselenocysteine and methylseleninic acids have also shown antimigratory effects in HeLa cells [24]. In addition, Tf-conjugated NPs encapsulating DOX demonstrate a higher inhibition of cell migration than free DOX in a DOX-resistant cell line [49].

As expected, Tf-decorated NPs have shown greater cellular uptake (Figure 7) than non-conjugated NPs in both sensitive and MDR cell lines, which may be assigned to the overexpression of TfR in the tumor cells, as these receptors bind to the Tf in the NP surface, resulting in endocytosis [40]. Other studies have also evidenced the increase in cellular uptake in different tumor cell lines for Tf-conjugated NPs in comparison with non-conjugated NPs [26,28,50,51]. Additionally, our results indicated that besides the increase in the cellular uptake rate, the Tf-decorated NPs also enhanced the intracellular retention (Figure 8), and this same behavior was observed in both sensitive and MDR cells. Therefore, the increase in cellular uptake and intracellular retention might support the higher antitumor activity and antimigratory effect shown by Tf-decorated NPs.

The Annexin V-FITC/PI assay was used to determine the type of cell death induced by the NPs. Apoptotic cells translocated phospholipid phosphatidylserine from the inner to the outer leaflet surface of the plasma membrane, and this externalization occurs in the earlier stages of apoptosis. Annexin V-FITC has a high affinity, and conjugates to phospholipid phosphatidylserine. Our results showed (Figure 9) that Tf-decorated NPs were the only treatment able to increase the number of apoptotic events in sensitive and MDR cells. These findings supported the 2D and 3D cytotoxicity results, which evidenced a higher cytotoxic activity of the Tf-conjugated NPs. Moreover, the enhancement in the apoptosis induction by the Tf-decorated NPs can be related to their higher cell internalization. Additionally, these data positively suggest that one of the cell death mechanisms induced by Tf-decorated NPs was apoptosis. Similarly, Li and coworkers [52] reported that Tf-conjugated NPs encapsulating piperine induced more apoptotic events than the non-conjugated counterparts, with the enhanced apoptosis being associated to the higher uptake and intracellular concentration.

In the cell cycle studies (Figure 10), the Tf-ACAT-Se-PLGA-NPs arrested the cell cycle in sub-G1 and G2/M in HeLa and in S and G2/M in NCI/ADR-RES cells. Considering that ACAT-Se is a thymidine analogue, we expect that its mechanism can either be mediated by enzyme inhibition or by replacing endogenous nucleoside species as substrates, leading to DNA and RNA damage and interference with DNA methylation [53]. In this sense, the majority of nucleoside analogues arrest the cell cycle in the S phase, when the DNA replication occurs [54]. In addition, the accumulation of cells in the sub-G1 phase is considered a biomarker for DNA damage, and can be related to the presence of apoptosis, confirming the findings achieved in the apoptosis experiments [55]. Ecker and coworkers [56] also studied thymidine analogues modified with selenium and observed a similar cellular mechanism; all three Se derivatives evaluated showed cell cycle arrest in the S and sub-G1 phase. Despite nucleoside analogues not usually arresting the cell cycle in the G2/M phase, Azuma and coworkers [57] evidenced that an analog of deoxycytidine (2′-C-cyano-2′-deoxy-1-β-D-arabino-pentofuranosylcytosine) presented cell cycle arrest in the G2/M phase. The authors verified that the compound showed an ability to induce DNA strand breaks, which were considered responsible for the cell cycle arrest in the G2/M phase.

## 5. Conclusions

In this study, we developed Tf-decorated ACAT-Se-loaded NPs in order to propose a novel targeted formulation for antitumor therapy, especially against MDR tumor cells. In this regard, our results showed a significant increase in cytotoxicity after the conjugation of Tf on the NPs’ surface, in all sensitive tumor cells studied as well as in the MDR tumor cell line, in both 2D and 3D models. The Tf-ACAT-Se-PLGA-NPs also displayed greater antimigrating activity than the nontargeted NPs, and was the only treatment able to induce apoptosis in the MDR cells. Finally, the improvements in the antitumor potential evidenced after the conjugation of Tf might be related to the higher cellular uptake and intracellular retention achieved by this NP formulation. Altogether, our findings suggested that the Tf-decorated ACAT-Se-loaded NPs proposed in this study are promising to increase the efficacy of antitumor therapy and overcome MDR. Further in vivo studies are necessary to confirm this premise.

## Figures and Tables

**Figure 1 nanomaterials-13-02306-f001:**
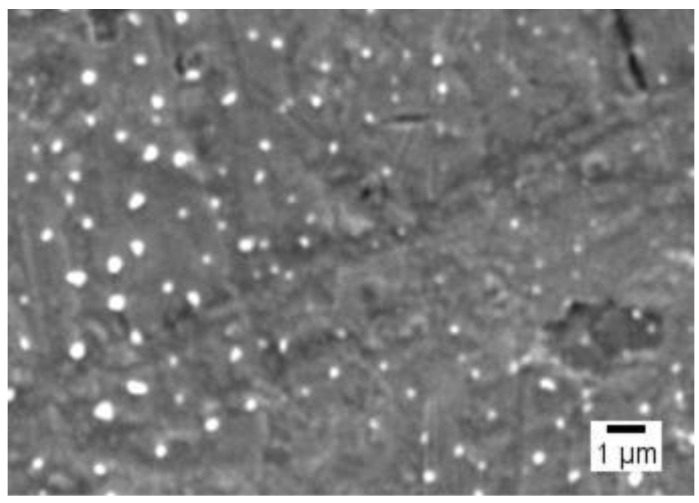
Tf-ACAT-Se-PLGA-NPs’ morphology by SEM.

**Figure 2 nanomaterials-13-02306-f002:**
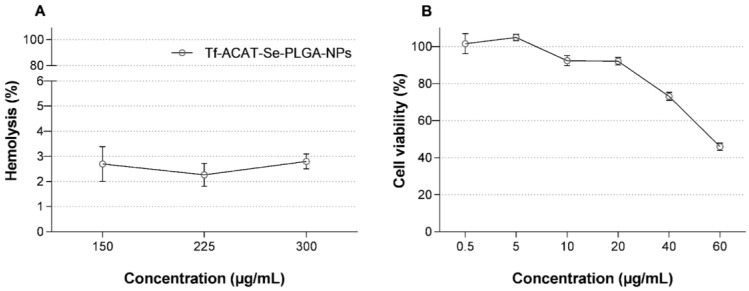
In vitro biocompatibility of Tf-ACAT-Se-PLGA-NPs. (**A**) Hemocompatibility study after incubation with human erythrocytes. (**B**) Cell biocompatibility using human mononuclear cells of peripheral blood (PBMCs) via MTT assay. Each value represents mean ± SE of three experiments.

**Figure 3 nanomaterials-13-02306-f003:**
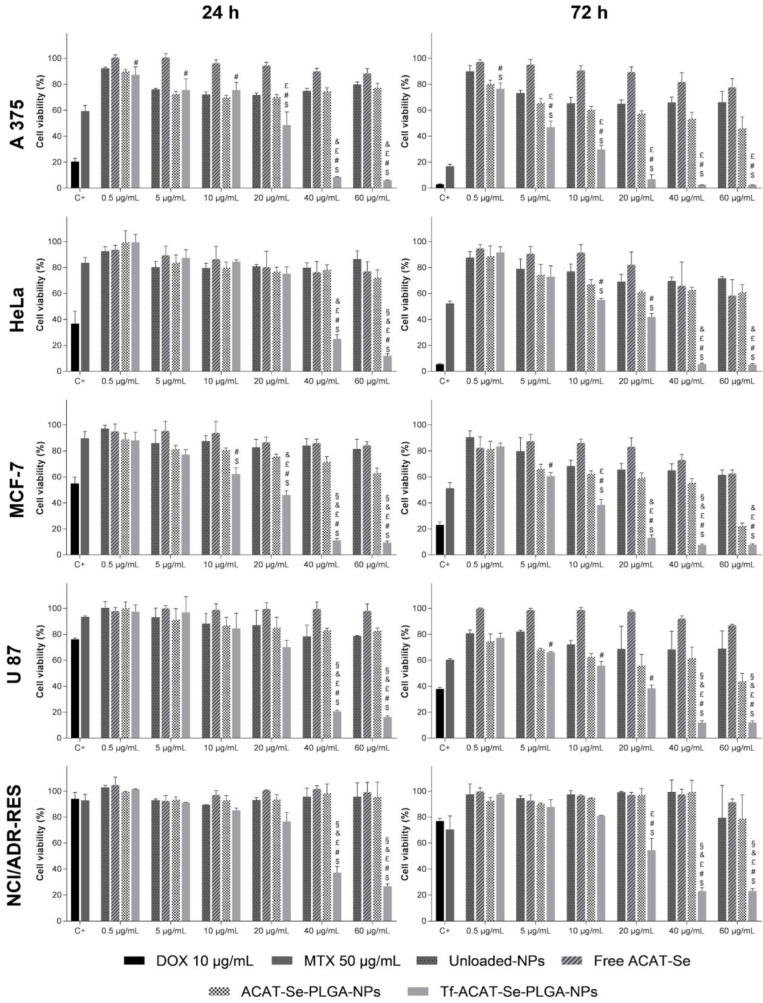
Cytotoxic effects of varying concentrations of free ACAT-Se, ACAT-Se-PLGA-NPs and Tf-ACAT-Se-PLGA-NPs on the survival rates of MCF-7, A375, HeLa, U87 and NCI/ADR-Res cell lines by using the MTT assay. Each value represents mean ± SE of three independent experiments. Statistical analyses were performed using ANOVA followed by the Student–Newman–Keuls multiple comparison test. ^$^ Significant difference from unloaded NPs, ^#^ significant difference from free ACAT-Se, ^£^ significant difference from ACAT-Se-PLGA-NPs, ^&^ significant difference from MTX, and ^§^ significant difference from DOX.

**Figure 4 nanomaterials-13-02306-f004:**
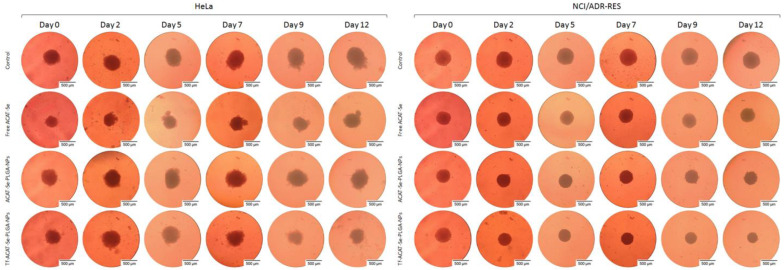
Representative images of spheroids treated with free ACAT-Se, ACAT-Se-PLGA-NPs or Tf-ACAT-Se-PLGA-NPs at 60 µg/mL (HeLa) or 120 µg/mL (NCI/ADR-RES). Images were obtained using inverted microscope at day 0 (before treatment), and after exposure to the treatments for 2, 5, 7, 9 and 12 days.

**Figure 5 nanomaterials-13-02306-f005:**
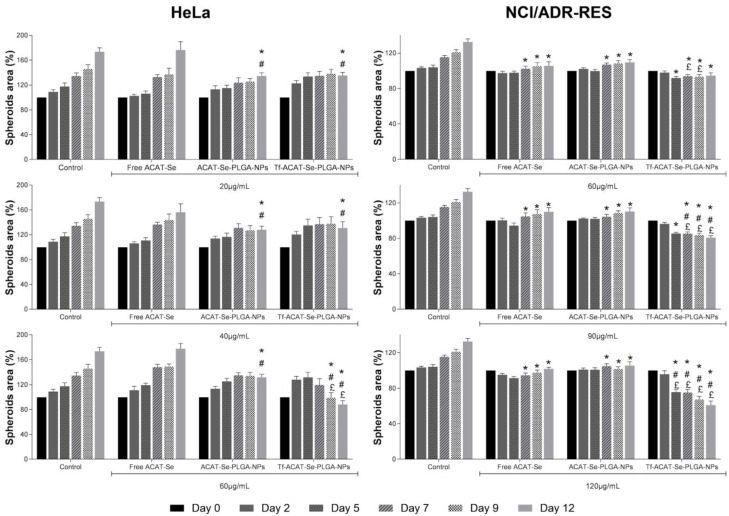
Cytotoxicity of free ACAT-Se, ACAT-Se-PLGA-NPs or Tf-ACAT-Se-PLGA-NPs against HeLa and NCI/ADR-RES spheroids. Spheroid area percentage (%) determined in comparison to day 0, which was set as 100%. The graphic y axis was set at 200 and 140%, for HeLa and NCI/ADR-RES, respectively, to better fit the spheroid growth. Statistical analyses were performed using ANOVA followed by the Student–Newman–Keuls multiple comparison test. * significant difference from control, ^#^ significant difference from free ACAT-Se, and ^£^ significant difference from ACAT-Se-PLGA-NPs.

**Figure 6 nanomaterials-13-02306-f006:**
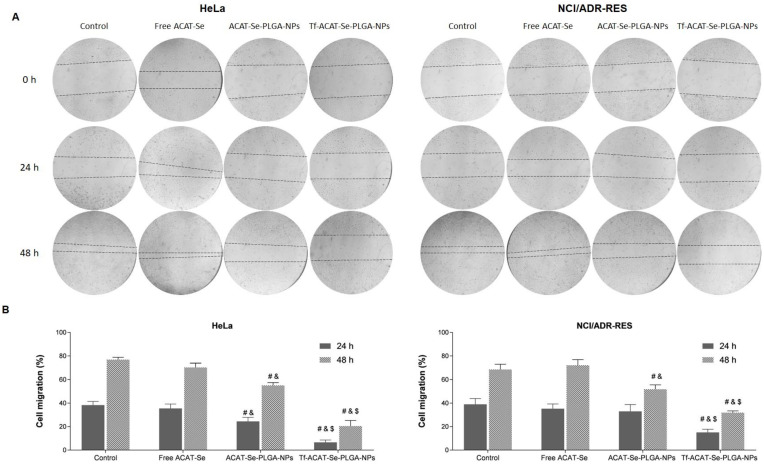
Effects of ACAT-Se on the motility of HeLa and NCI/ADR-Res cells. (**A**) Images obtained using inverted microscope at time points 0, 24 and 48 h. (**B**) Cell migration percentages after 24 and 48 h treatments. Statistical analyses were performed using ANOVA followed by the Student–Newman–Keuls multiple comparison test. ^#^ is different from control, ^&^ is different from free ACAT-Se and ^$^ is different from ACAT-Se-PLGA-NPs (*p* < 0.05).

**Figure 7 nanomaterials-13-02306-f007:**
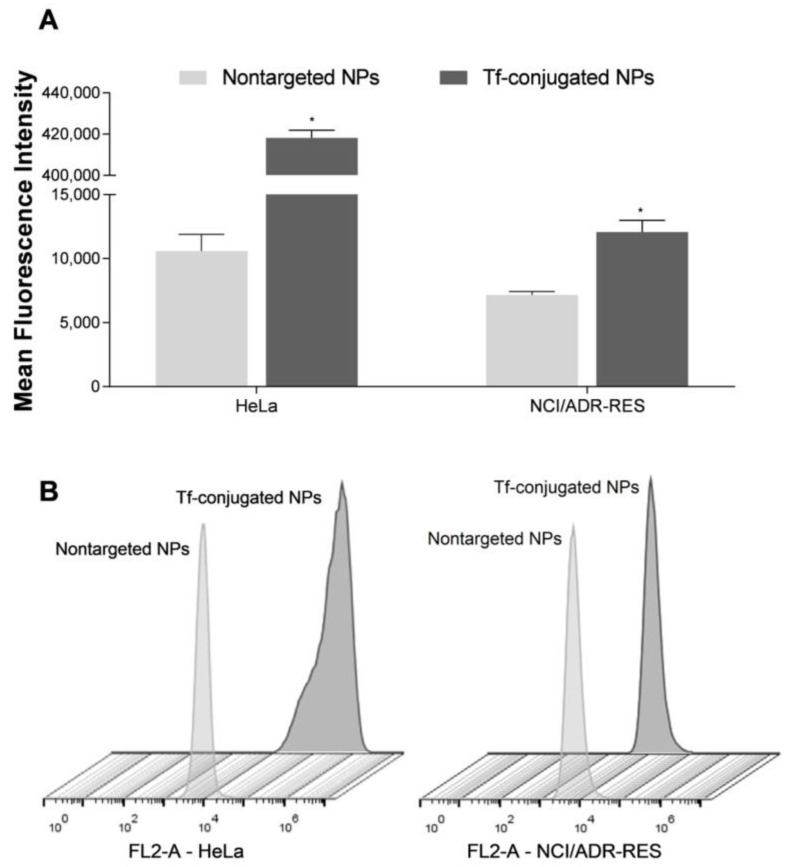
Cell uptake of nontargeted NPs and Tf-conjugated NPs in HeLa and NCI/ADR-RES cells. (**A**) Mean fluorescence intensity determined using flow cytometry. (**B**) Flow cytometry profiles of cellular uptake. Results are expressed as mean ± SE of three independent experiments. Statistical analyses were performed using ANOVA followed by the Student–Newman–Keuls multiple comparison test. * Significant difference from nontargeted NPs (*p* < 0.05).

**Figure 8 nanomaterials-13-02306-f008:**
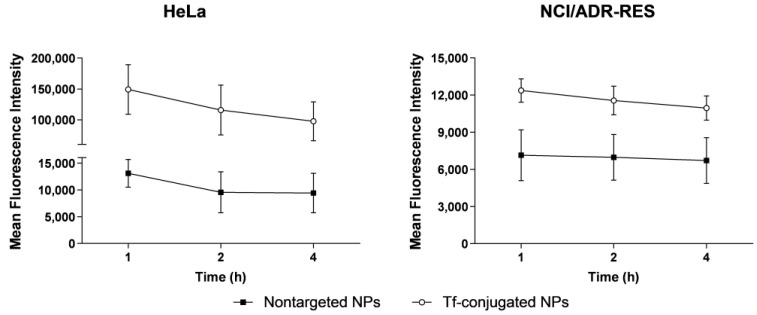
Intracellular retention of nontargeted NPs and Tf-conjugated NPs in HeLa and NCI/ADR-RES cells. Results are expressed as mean ± SE of three independent experiments.

**Figure 9 nanomaterials-13-02306-f009:**
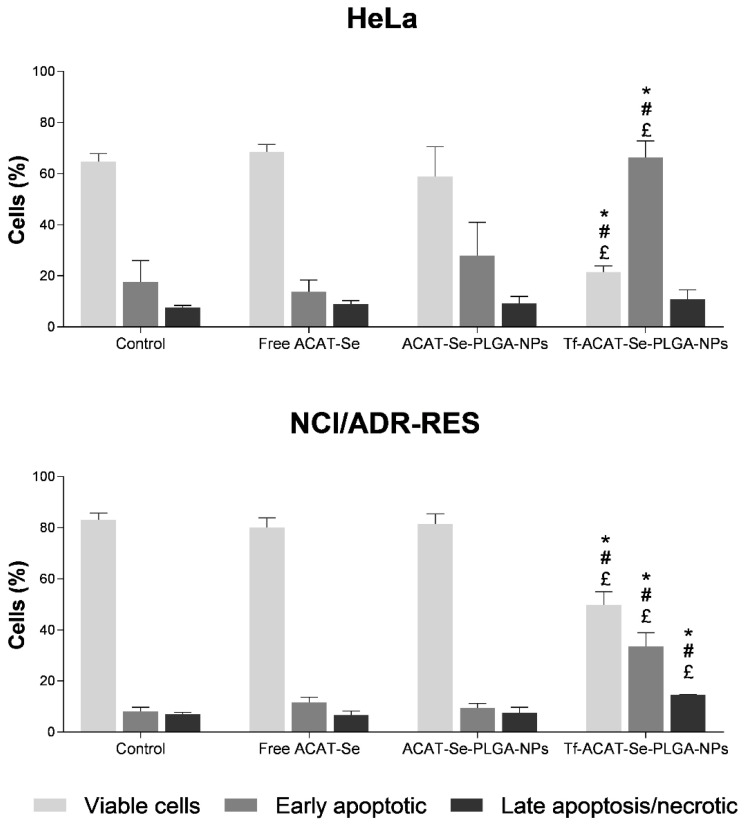
Cell apoptosis induced by free ACAT-Se, ACAT-Se-PLGA-NPs or Tf-ACAT-Se-PLGA-NPs in HeLa and NCI/ADR-RES cells. The results are expressed as mean ± SE of three independent experiments by means of the percentage of viable, early apoptotic and late apoptotic/necrotic cells after 24 h incubation with the treatments. Statistical analyses were performed using ANOVA followed by the Student–Newman–Keuls multiple comparison test. * significant difference from control, ^#^ significant difference from free ACAT-Se, and ^£^ significant difference from ACAT-Se-PLGA-NPs.

**Figure 10 nanomaterials-13-02306-f010:**
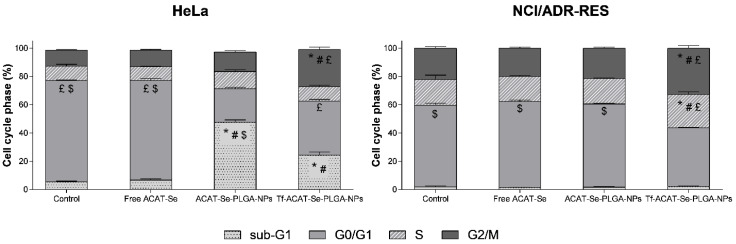
Cell cycle analysis for HeLa and NCI/ADR-RES cells treated with free ACAT-Se, ACAT-Se-PLGA-NPs or Tf-ACAT-Se-PLGA. The results are expressed as mean ± SE of three independent experiments by means of the percentage of cells in the sub-G1, G0/G1, S or G2M phase. Statistical analyses were performed using ANOVA followed by the Student–Newman–Keuls multiple comparison test. * significant difference from control, ^#^ significant difference from free ACAT-Se, ^£^ significant difference from ACAT-Se-PLGA-NPs, and ^$^ significant difference from Tf-ACAT-Se-PLGA-NPs.

**Table 1 nanomaterials-13-02306-t001:** IC_50_ (µg/mL) and SI values for Tf-ACAT-Se-PLGA-NPs after 24 h of treatment.

Tf-ACAT-Se-PLGA-NPs		
	IC_50_	SI
A375	14.38	4.17
HeLa	31.64	1.90
MCF-7	14.40	4.17
U-87	27.04	2.22
NCI/ADR-RES	31.48	1.91
PBMCs	60.02 *	-

* Estimated IC_50_ based on dose–response curve.

## Data Availability

The data presented in this study are available on request from the corresponding author.

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
