# Peer review of "Transferrin-Decorated PLGA Nanoparticles Loaded with an Organoselenium Compound as an Innovative Approach to Sensitize MDR Tumor Cells: An In Vitro Study Using 2D and 3D Cell Models"

_nanomaterials, 2023, doi:10.3390/nano13162306_

Round 1
Reviewer 1 Report
Many data in this manuscript have confirmed that Tf-ACAT-Se-PLAG-NPS has the antitumor’s ability.
1:ACAT-Se, ACAT-Se-PLGA-NPS and Tf-ACAT-Se-PLAG-NPS treats Hela or NCI/ADR-RES and harvest the cell lysis, western blot caspase3 antibody.
2:In vivo mouse tumor model, three groups of ACAT-Se, ACAT-Se-PLGA-NPS and Tf-ACAT-Se-PLAG-NPS treatment, test tumor size and western blot apoptosis related proteins.
Author Response
The authors are grateful for the reviewer’s comments, improving the quality of the article. The suggestions are included in the new version of the manuscript as it can be seen in the answers to each of the comments. The corrections and changes included in the revised version of the article are identified in red, here and in the text of the manuscript.

Reviewer 2 Report
The manuscript designed transferrin (Tf)-conjugated PLGA nanoparticles (NPs) containing a new organoselenium compound as an alternative to enhance the efficacy of cancer therapy and sensitize MDR tumor cells. Although the proposed idea was demonstrated, this concept and strategy are relatively uneventful, and the conclusions are not adequately supported by the experimental data.
1. The author claimed that they developed a new organoselenium compound, but the ACAT-Se has been reported before.
2. The SEM images (Figure 1) exhibited an uneven size distribution of nanoparticles, and images with higher resolution should be provided. Other characterizations such as TEM should be provided.
3. The corresponding images of particle size, PDI and ZP should be provided.
4. For in vitro release studies, why use free ACAT-Se as the control? How to determine the release percentage of free ACAT-Se?
5. How was ACAT-Se released from Tf-ACAT-Se-PLGA-NPs? If pH-sensitive, why not perform this experiment at different pH?
6. In in vitro biocompatibility studies, the cell viability was lower than 80% when cells treated with Tf-ACAT-Se-PLGA-NPs at the concentration of 40 μg/mL. The author can’t say Tf-ACAT-Se-PLGA-NPs displayed negligible cytotoxicity in PBMCs.
7. The author described the ACAT-Se presented a promising in vitro antitumor activity, why free ACAT-Se displayed negligible antitumor activity?
8. What is the mechanism of ACAT-Se for tumor therapy? Why it possessed radical scavenging activity?
9. For in vitro antitumor activity assays using 3D cell models, the live/dead cell staining should be performed to evaluate the antitumor activity. And the scale bar should be provided.
There are many errors and grammatical mistakes in the manuscript.
Author Response

(The authors gave the same response as above.)

Reviewer 3 Report
This manuscript presents the study of transferrin-decorated nanoparticles loaded with an organoselenium derivative. The concept of using transferrin as a targeting agent is not innovative. The paper suffers from a lack of details about the methodology and the products that were used and some controls are missing as well.
It seems that part of the results (esp., results for unconjugated particles) comes from a former paper of the same team (ref 12 in the paper).
It appears that the free drug has no effect on cells (figures 3 and 5), therefore, why want to use this drug that shows very little potential as a cytotoxic drug? What is the need for this compound to be encapsulated? Has this drug been administered in animals?
The authors reports that the drug in Tf modified particles does not have the same mode of action than the free drug. This aspect should be more discussed.
It would be useful to have some information regarding the drug, for instance, its aqueous solubility and solubility in different common solvent.
Several information in the material and methods are missing.
The nature of the components of the nanoparticles is not given.
Especially, what was the PLGA used? MM? Lactic-Glycolic ratio? End group? Provider?
The preparation methods refer to an article by the same authors (ref 12). However, in the cited paper, several types of particles were made. A brief description of the process used for this paper should be given, more specifically regarding the surfactants that were used. In ref. 12, it is mentioned that the surfactant mass was higher than the polymer: PLGA: 0.05g – SPAN: 0.08 and pluronic: 0.150g. Why use this amount of surfactant? Was it all eliminated during the purification process. In the present study, was the pH-sensitive adjuvant 77KL added?
In the discussion of the current paper, it is claimed that the particles are pH-Sensitive, this aspect is not demonstrated here, this data should be included.
What was medium in which the Tf was added, buffer? pH?
How were the particles purified?
A priori, the NP were not lyophilized, then how were the particles weighted for the determination of the encapsulation efficiency? Is some loss in polymer mass taken into account in the calculation? How?
Regarding the cytotoxicity and efficiency studies, there should be a control made of unloaded decorated particles to assess that the effect is due to the presence of the drug and not the conjugation of Tf on the particles’ surface.
In the results section:
Decimals are not necessary when mentioning the size as the measurement is not precise enough.
In the characterization of the particles part, size of the different batches should be given in a table.
Figure 1 is too small to get a real idea about the particles shape and size, please enlarge.
Regarding particle characterization data, this study should not be compared to the data from the previous article (ref 12) but to the actual batches made during this study.
The biocompatibility and hemolysis studies, the table 1 should be completed with the results of the free drug, drug-loaded unmodified particles, and the unloaded modified particles.
In the antitumour activity study, were the unloaded particles surface modified or not?
For the internalization study, fluorescence approach and FACS analysis were used. However, fluorescence is depending on the surrounding environment and the encapsulation rate of the probe. To be able to compare both formulations, a standardization should be used, or the authors should bring the proof that the fluorescence associated with the two batches was the same.
Furthermore, some pictures should be added to assess whether the fluorescence corresponds to internalized fluorescence or merely associated to the cell surface.
Moreover, the demonstration of the role of Tf modification should be completed with a competition study where free Tf is added in the medium and blocks the interaction of the modified particles.
For the release study (figure S1), the curve seems to reach a plateau, when is the totality of the drug released? Was a mass (amount of drug still encapsulated in the polymer matrix at the time of the end of the study) balance done to make sure that the remaining drug is still in the system?
What is the release profile of the non-modified particles?
For the study about apoptosis rate, does that really bring new information compared to the MTT study? This seems redundant.
In the discussion, the authors claim that the better results obtained with the modified nanoparticles are based on a better stability of the drug. However, this is not demonstrated in the paper, this should be added. What is the stability problem linked to that molecule?
Regarding the absence of protein corona, the results are quite surprising and not in agreement with the published literature. This should be discussed. From a general standpoint, the interaction with protein is more influenced by the hydrophobicity of the system than the zeta potential.
Editing remarks:
Check that all acronyms are defined.
Add the acronym signification in figure S4.
What are the units in table S1?
In the discussion, please remind the figure linked to the data that are discussed, it would make it easier to follow.
In the M&M, many spaces are missing between words, especially in equations “??????????????????“
Some English mistakes were noticed and should be checked, for example:
“The rate of Tf conjugation were evaluated…”
“The cell viability was determinate by MTT..”
“increased the number of apoptotic events in compassion to all other treatments…”
Author Response

(The authors gave the same response as above.)

Reviewer 4 Report
In the manuscript entitled "Transferrin-decorated PLGA nanoparticles loaded with a new organoselenium compound as an innovative approach to sensitize MDR tumor cells: an in vitro study using 2D and 3D cell models" the author describe the results obtained using a drug delivery system against MDR tumor cells in 2D and 3D cell models.
The study was well conducted and the results are well reported, with good results obtained for the new systems generated.
A small suggestion can be made. In the material and methods section, some of the methods where not describe. It could be useful for the reader to have at least a brief description of the methods in the manuscript.
Author Response

(The authors gave the same response as above.)

Round 2
Reviewer 2 Report
All comments addressed. No more concerns.